## BRIEF COMMUNICATION
# Oxford Nanopore R10.4 long-read sequencing enables the generation of near-finished bacterial genomes from pure cultures and metagenomes without short-read or reference polishing

Mantas Sereika [1,4], Rasmus Hansen Kirkegaard [1,2,4], Søren Michael Karst [1], Thomas Yssing Michaelsen[1], Emil Aarre Sørensen [1], Rasmus Dam Wollenberg[3] and Mads Albertsen [1]✉

**Long-read Oxford Nanopore sequencing has democratized microbial genome sequencing and enables the recovery of highly contiguous microbial genomes from isolates or metagenomes. However, to obtain near-finished genomes it has been necessary to include short-read polishing to correct insertions and deletions derived from homopolymer regions. Here, we show that Oxford Nanopore R10.4 can be used to generate near-finished microbial genomes from isolates or metagenomes without short-read or reference polishing.**

Bacteria live in almost every environment on Earth and the global microbial diversity is estimated to entail more than $10^{12}$ species[1]. To obtain representative genomes, either sequencing of pure cultures or recovery of genomes directly from metagenomes are often used[2–4]. High-throughput short-read sequencing has for many years been the method of choice[5,6] but it fails to resolve repeat regions larger than the insert size of the library[7]. This is especially problematic in metagenome samples, in which related species or strains often contain long sequences of near-identical DNA. More recently, long-read sequencing has emerged as the method of choice for both pure culture genomes[8–10] and metagenomes[11–15]. PacBio HiFi reads combine low error rates with relatively long reads and generate near-finished microbial genomes from pure cultures or metagenomes[16–18]. Despite the very high-quality raw data, the relatively high cost per base remains an economic hindrance for many research projects. A widely used alternative is Oxford Nanopore sequencing, which offers low-cost long-read data. However, numerous studies have shown that despite vast improvements in raw error rates, assembly consensus sequences still contain insertions and deletions in homopolymers (indels) that often cause frameshift errors during gene calling[19–21]. A commonly adopted solution has been to include short-read data for post-assembly error correction[15,22], although it increases the cost and complexity overhead. Another solution has been to apply reference-based polishing to correct frameshift errors[23–25] but, although this provides a practical solution that enables gene calling, it does not provide true near-finished genomes. Finished microbial genomes, as defined by Bowers et al. 2017 in the MIMAG (minimum information about a metagenome-assembled genome) standard[26], are genomes that have "…a single, validated, contiguous sequence per replicon, without

gaps or ambiguities" and "a consensus error rate equivalent to Q50 or better". This is difficult to achieve even with multiple sequencing technologies on pure cultures[19] and metagenome-assembled genomes (MAGs)[27]. However, the second-highest quality tier, high quality, can be achieved despite large amounts of frameshift errors, which can have large implications for downstream analysis[20]. Hence, we here introduce the term 'near-finished' genome and define it as a high-quality genome for which short-read polishing is not expected to significantly improve the consensus sequence.

We first evaluated the ability to obtain near-finished microbial genomes from Oxford Nanopore R9.4.1 and R10.4 data through sequencing of the ZymoBIOMICS HMW (high molecular weight) DNA Standard D6322 (Zymo mock) consisting of seven bacterial species and one fungus. A single PromethION R10.4 flowcell generated 52.3 Gbp of data with a modal read accuracy of 99% (Fig. 1a and Supplementary Table 1). In contrast to the R9.4.1 data, we do not see any significant improvement in the assembly quality for R10.4 by the addition of Illumina polishing (Fig. 1c and Supplementary Fig. 1). This indicates that near-finished microbial reference genomes can be obtained from R10.4 data alone at a coverage of approximately 40-fold (Supplementary Table 2). The improvement in assembly accuracy from R9.4.1 to R10.4 is largely due to an improved ability to call homopolymers (Fig. 1b and Supplementary Figs. 2 and 3). Even though there is some nucleotide-specific variation in homopolymer calling accuracy at lengths 8 and 9 on a read level (especially with cytosines), on a genome consensus level the vast majority of homopolymers are correctly resolved up to a length of <11 bp in R10.4 data (Supplementary Fig. 4). In general, long homopolymers are very rare in bacteria[21], and by analyzing complete genomes from 1,598 different genera (Supplementary Fig. 5) we found only 18 genomes (1%) with long homopolymers (>10), at a rate of more than 1 per 100,000 bp (theoretical Q50 limit).

To assess the performance of state-of-the-art sequencing technologies in recovering near-finished microbial genomes from metagenomes we sequenced activated sludge from an anaerobic digester using single runs of Illumina MiSeq 2 × 300 bp, PacBio HiFi, and Oxford Nanopore R9.4.1 and R10.4. Despite being the same sample, direct comparisons are difficult because the additional size selection of the PacBio HiFi dataset both increased the read length

[1]Center for Microbial Communities, Aalborg University, Aalborg, Denmark. [2]Joint Microbiome Facility, University of Vienna, Vienna, Austria. [3]DNASense ApS, Aalborg, Denmark. [4]These authors contributed equally: Mantas Sereika, Rasmus Hansen Kirkegaard. ✉e-mail: ma@bio.aau.dk

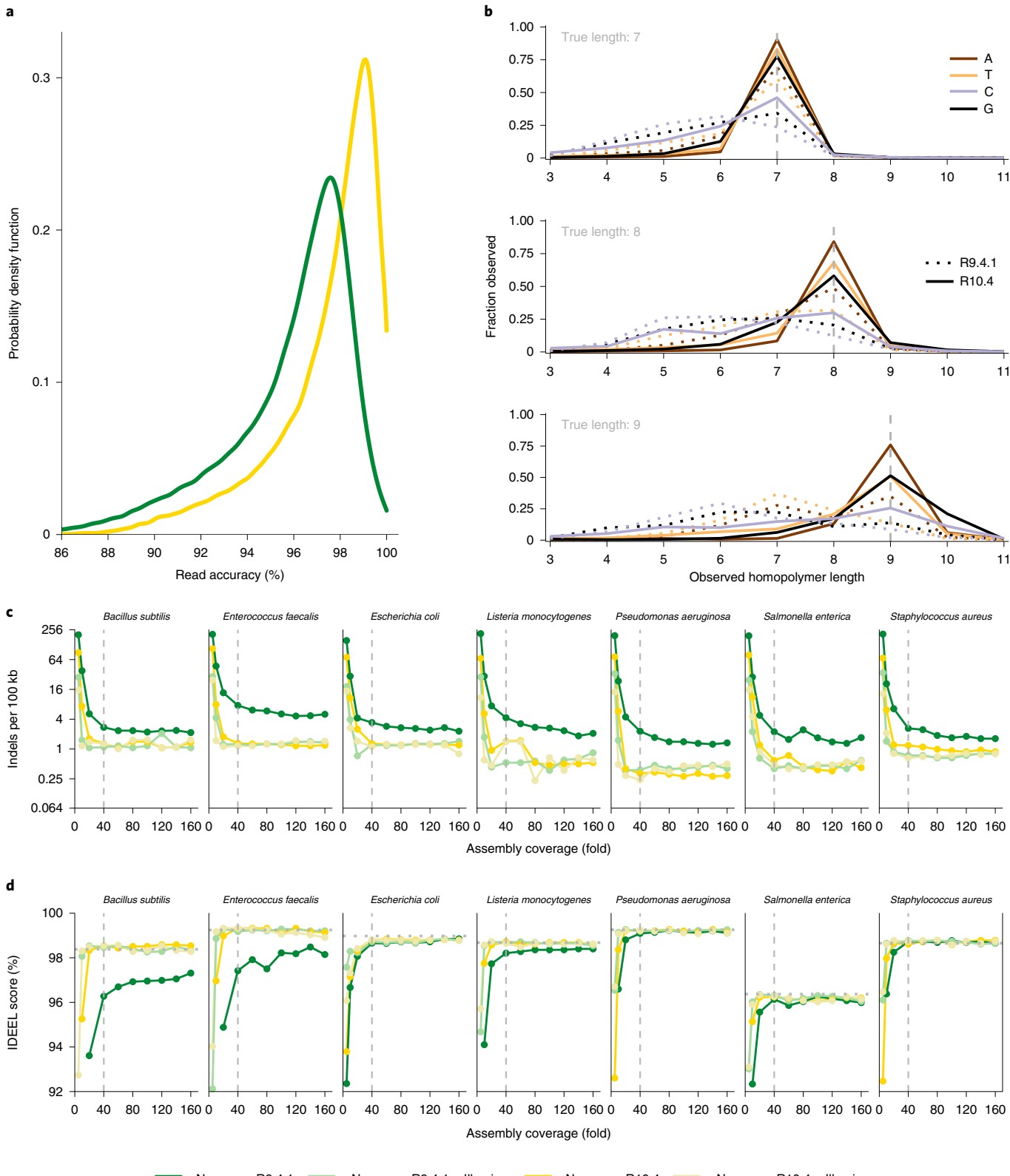

**Fig. 1 | Sequencing and assembly statistics for the Zymo mock bacterial species (*n* = 7). a**, Observed raw read accuracies measured through read-mapping. **b**, Observed homopolymer length of raw reads compared with the reference genomes (see Supplementary Figs. 2 and 3 for a complete overview). **c**, Observed indels of de novo assemblies per 100 kbp at different coverage levels, with and without Illumina polishing. Note that the reference genomes available for the Zymo mock are not identical to the sequenced strains (Supplementary Table 3). **d**, IDEEL[28] score, calculated as the proportion of predicted proteins that are ≥95% the length of their best-matching known protein in a database[19]. The dotted line represents the IDEEL score for the reference genome, while the dashed lines mark a 40-fold coverage cut-off.

**Table 1 | Sequencing and assembly statistics for the anaerobic digester sample using different technologies and approaches**

|  | Illumina MiSeq | R9.4.1 / + Illumina | R10.4 / + Illumina | PacBio HiFi/ + Illumina |
|---|---|---|---|---|
| **Total yield (Gbp)** | 13 | 35 | 14 | 15 |
| **Read N50 (kbp)** | 0.3 | 5.9 | 5.6 | 15.4 |
| **Observed modal read accuracy (%)[a]** | 100 | 96.77 | 98.11 | 99.93 |
| **Assembly size (Mbp)** | 409 | 754 | 379 | 606 |
| **Contigs (>1kbp)** | 145,976 | 24,680 | 21,585 | 8,989 |
| **Circular contigs (>0.5 Mbp)** | 0 | 7 | 3 | 9 |
| **Contig N50 (kbp)** | 3.5 | 79.9 | 40.1 | 172.5 |
| **Reads mapped to contigs (%)** | 88.1 | 93.5 | 95.4 | 95.2 |
| **HQ MAGs** | 8 | 64/86 | 34/36 | 74/77 |
| **MQ MAGs** | 83 | 114/95 | 65/67 | 72/68 |
| **No. of contigs per HQ MAG (median)** | 184 | 15/16 | 21/21 | 9/10 |
| **Single-contig HQ MAGs** | 0 | 2/3 | 1/1 | 3/3 |
| **Mapped reads in HQ MAGs (%)** | 16 | 46/49 | 39/40 | 48/44 |
| **Cost (US$)[b]** | 1,200 | 811/2,011 | 811/2,011 | 4,420/5,620 |
| **Cost per HQ MAG (US$)** | 150 | 13/23 | 24/56 | 60/73 |

HQ, high quality. xx/xx, short-read unpolished/polished assemblies, relevant only for MAG quality statistics because the overall assembly statistics are identical. [a]Observed read accuracies calculated from read mappings to an Illumina-polished PacBio HiFi assembly. [b]The expenses encountered at the time of conducting the experiments. This may differ for other research groups.

HiFi data (N50 read length 6 kbp versus 15 kbp) only small differences in bin fragmentation were observed, as compared with the Illumina-based results (Table 1 and Supplementary Fig. 8).

All long-read methods produce high numbers of high-quality MAGs, which capture 39–49% of all reads (Table 1). Nanopore R9.4.1 is able to produce high-quality MAGs as a standalone technology, but Illumina polishing increases the number of high-quality MAGs from 64 to 86. For Nanopore R10.4, Illumina polishing increases the number of high-quality MAGs from 34 to 36. Using the IDEEL score[19] (Supplementary Fig. 9) as a relative measurement for improvement in genome consensus quality, Illumina polishing results in minor improvements for Nanopore R10.4 above a coverage of 40, and the Nanopore R10.4 is in the same IDEEL range as PacBio HiFi MAGs. As with sequencing of the Zymo mock, the difference from R9.4.1 to R10.4 is largely due to the significantly better accuracy in homopolymers for lengths up to 10 (Supplementary Fig. 4).

Since its introduction as an early access program in 2014 Oxford Nanopore sequencing technology has democratized sequencing and enabled more laboratories and classrooms to engage in microbial genome sequencing. However, for the generation of high-quality genomes, additional short-read polishing has been essential, given that indels in homopolymer regions cause fragmented gene calls. The additional sequencing requirements have been one of the barriers to widespread uptake. Here, we show that Oxford Nanopore R10.4 enables the generation of near-finished microbial genomes from pure cultures or metagenomes at coverages of 40-fold without short-read polishing. Although homopolymers of 10 or more bases will probably still be problematic, they constitute a minor part of microbial genomes (Supplementary Fig. 5).

For genome recovery from metagenomes, low-coverage bins (<40-fold) do need Illumina polishing to achieve a quality comparable to PacBio HiFi. Hence, in some cases the most economic option could be Nanopore R9.4.1 supplemented with short-read sequencing, given that the throughput is currently at least twofold higher on R9.4.1 compared with R10.4 and no difference is seen between the methods after Illumina short-read polishing.

## Online content

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

## Methods

**Sampling.** Sludge biomass was sampled from the anaerobic digester at Fredericia Wastewater Treatment Plant in Denmark (latitude 55.552219, longitude 9.722003) at multiple time points and stored as frozen 2 mL aliquots at −20 °C. For the Zymo sample, the ZymoBIOMICS HMW DNA Standard D6322 (Zymo Research) was used.

**DNA extraction.** DNA was extracted from the anaerobic digester sludge using the DNeasy PowerSoil Kit (Qiagen) following the manufacturer's protocol. The extracted DNA was then size selected using the SRE XS kit (Circulomics), according to the manufacturer's instructions, to deplete DNA fragments below 10 kbp.

**DNA QC.** DNA concentrations were determined using the Qubit dsDNA HS kit and were measured with a Qubit 3.0 fluorimeter (Thermo Fisher). DNA size distribution was determined using an Agilent 2200 Tapestation system with genomic screentapes (Agilent Technologies). DNA purity was determined using a NanoDrop One Spectrophotometer (Thermo Fisher).

**Oxford Nanopore DNA sequencing.** Library preparation was carried out using the ligation sequencing kits (Oxford Nanopore Technologies) SQK-LSK109 and SQK-LSK112 for sequencing on R.9.4.1 and the R.10.4 flowcells, respectively. Anaerobic digester and Zymo R.9.4.1 datasets were generated on a MinION Mk1B (Oxford Nanopore Technologies) device, while the Zymo R10.4 dataset was produced on a PromethION and the digester R10.4 read sequences were generated on a GridION using the MinKNOW v21.05.25 software (https://community.nanoporetech.com/downloads).

**Illumina DNA sequencing.** The anaerobic digester Illumina libraries were prepared using the Nextera DNA library preparation kit (Illumina), while the Zymo Mock sample was prepared with the NEB Next Ultra II DNA library prep kit for Illumina (New England Biolabs) following the manufacturer's protocols and sequenced using the Illumina MiSeq platform.

**PacBio HiFi sequencing.** A size-selected DNA sample was sent to the DNA Sequencing Center at Brigham Young University, Provo, Utah, USA. The DNA sample was fragmented with Megaruptor (Diagenode) to 15 kbp and size-selected (>10 kbp) using the Blue Pippin (Sage Science), and prepared for sequencing using the SMRTbell Express Template Preparation Kit 1.0 (PacBio) according to the manufacturer instructions. Sequencing was performed on the Sequel II system (PacBio) using the Sequel II Sequencing Kit 1.0 (PacBio) with the Sequel II SMRT Cell 8M (PacBio) for a 30 h data collection time.

**Read processing.** Illumina reads were trimmed for adapters using Cutadapt v. 1.16 (ref. [29]). The generated raw Nanopore data were basecalled in super-accurate mode using Guppy v. 5.0.16 (https://community.nanoporetech.com/downloads) with the dna_r9.4.1_450bps_sup.cfg model for R9.4.1 and the dna_r10.4_e8.1_sup.cfg model for R10.4 chemistry. Given that the R10.4 data were observed to feature concatemeric reads that might complicate the metagenome assembly step, the concatemers in R10.4 data were split by using the split_on_adapter command (five iterations) of duplex-tools v. 0.2.5 (https://github.com/nanoporetech/duplex-tools). Adapters for Nanopore reads were removed using Porechop v. 0.2.3 (ref. [30]), and reads with a lower length than 200 bp and a Phred quality score below 7 and 10 for R9.4.1 and R10.4 reads, respectively, were removed using NanoFilt v. 2.6.0 (ref. [31]). The CCS tool v. 6.0.0 (https://ccs.how/) was used with the PacBio sub-read data to produce HiFi reads. Read statistics were acquired via NanoPlot v. 1.24.0 (ref. [31]). Counterr v. 0.1 (https://github.com/dayzerodx/counterr) was used to assess homopolymer calling in reads.

Long- and short-read datasets for the Zymo Mock bacterial species were subsampled according to custom coverage profiles (range, 5–160) using Rasusa v. 0.3.0 (https://github.com/mbhall88/rasusa), with the notable exception of *Pseudomonas aeruginosa*, which featured a maximum coverage of 92 in the short-read dataset. *Saccharomyces cerevisiae* data were excluded from the Zymo Mock analysis due to insufficient coverage. Anaerobic digester R9.4.1 read data were subsampled using the command 'seqtk sample -s100 0.37' from seqtk v. 1.3 (https://github.com/lh3/seqtk).

**Read assembly and binning.** Long reads were assembled using Flye v. 2.9-b1768 (refs. [16,32]) with the '–meta' setting enabled and the '–nano-hq' option for assembling Nanopore reads, whereas the '–pacbio-hifi' and '–min-overlap 7500–read-error 0.01' options were used for assembling PacBio HiFi reads, given that it resulted in more high-quality MAGs than using the default settings. The polishing tools for the Nanopore-based assemblies consisted of Minimap2 v. 2.17 (ref. [33]), Racon v. 1.3.3 (used three times)[34], Medaka v. 1.4.4 (used twice, https://github.com/nanoporetech/medaka), and one round of Racon with Illumina reads. For the short-read assembly the trimmed Illumina reads were assembled using Megahit v. 1.1.4 (ref. [35]). Contigs shorter than 1 kbp were filtered out using Bioawk v. 1.0 (https://github.com/lh3/bioawk). The contig guanine and cytosine content was calculated using infoseq (v. 6.6.0.0, ref. [36]).

Automated binning was carried out using three binners: MetaBAT2 v. 2.12.1 (ref. [37]) with the '-s 500000' setting, MaxBin2 v. 2.2.7 (ref. [38]), and Vamb v. 3.0.2 (ref. [39]) with the '-o C–minfasta 500000' setting. To aid with the binning process, contig coverage profiles from different sequencer datasets (Supplementary Table 1) as well as contig coverage by nine additional time-series Illumina datasets of the same anaerobic digester (Supplementary Table 4) were provided as input to the three binners. The binning output of different tools was then integrated and refined using DAS Tool v. 1.1.2 (ref. [40]). CoverM v. 0.6.1 (https://github.com/wwood/CoverM) was applied to calculate the bin coverage (using the '-m mean' setting) and the relative abundance ('-m relative_abundance'). A general overview of the processing of the sludge metagenomic data is presented in Supplementary Fig. 10.

**Assembly processing.** The completeness and contamination of the genome bins were estimated using CheckM v. 1.1.2 (ref. [41]). The bins were classified using GDTB-Tk v. 1.5.0 (ref. [42]) and the R202 database. Protein sequences were predicted using Prodigal v. 2.6.3 (ref. [43]) with the 'p meta' setting, while the ribosomal RNA genes were predicted using Barrnap v. 0.9 (https://github.com/tseemann/barrnap) and the transfer RNA predictions were made using tRNAscan-SE v. 2.0.5 (ref. [44]). Bin quality was determined following the Genomic Standards Consortium guidelines, in which a MAG of high quality has genome completeness of more than 90%, contamination of less than 5%, at least 18 distinct tRNA genes, and an occurrence of at least once of the 5S, 16S and 23S rRNA genes[26]. MAGs with completeness above 50% and contamination below 10% were classified as medium quality, while low-quality MAGs featured completeness below 50% and contamination below 10%. MAGs with contamination estimates higher than 10% were classified as contaminated.

Illumina reads were mapped to the assemblies using Bowtie2 v. 2.4.2 (ref. [45]) with the '–very-sensitive-local' setting. The mapping was converted to BAM and sorted using SAMtools v. 1.9 (ref. [46]). The single-nucleotide polymorphism rate was then calculated using CMseq v. 1.0.3 (ref. [6]) from the mapping using poly.py script with the '–mincov 10–minqual 30' setting.

Bins were clustered using dRep v. 2.6.2 (ref. [47]) with the '-comp 50 -con 10 -sa 0.95' setting. Only the bins that featured higher coverage than 10 in their respective sequencing platform and a higher Illumina read coverage than 5 for bins from the hybrid approach were included in downstream analysis. The IDEEL test was used to infer the level of protein truncations in the bins and was applied to provide a relative measurement of improvement in genome consensus quality via short-read polishing[20,28]. In brief, the predicted protein sequences from clustered bins and Zymo assemblies were searched against the UniProt TrEMBL[48] database (release 2021_01) using Diamond v. 2.0.6 (ref. [49]). Query matches, which were not present in all datasets, were omitted to reduce noise. The IDEEL scores (estimated fraction of full-length protein sequences) were assigned as described previously[19], where query-to-reference length ratios of more than 0.95 were counted as full-length protein sequences.

QUAST v. 4.6.3 (ref. [50]) was applied on the Zymo assemblies and the clustered bins that had a single-nucleotide polymorphism rate less than 0.5% to determine the mismatch and indels metrics. Cases with the QUAST parameters genome fraction less than 75% and unaligned length more than 250 kbp were omitted to reduce noise. For homopolymer analysis, the clustered bins were mapped to each other using the asm5 mode of Minimap2, and Counterr was used on the mapping files to determine the homopolymer calling errors. For QUAST and Counterr, Illumina-polished PacBio HiFi bins were used as reference sequences. FastANI v. 1.33 (ref. [51]) was used to calculate identity scores between Zymo assemblies and the Zymo reference sequences. The Zymo mock reference genome sequences, which were used as a substitute for PacBio HiFi, were obtained from a link in the accompanying instruction manual to the ZymoBIOMICS HMW DNA Standard Catalog No. D6332 (https://s3.amazonaws.com/zymo-files/BioPool/D6322.refseq.zip).

**Genome database analysis.** Archeal and bacterial genomes from the National Center for Biotechnology Information (NCBI) Reference Sequence (RefSeq) genome database were downloaded using ncbi-genome-download v. 0.3.0 (https://github.com/kblin/ncbi-genome-download, downloaded on 24 November 2021) with the '–assembly-levels complete' option. Genomes were subsampled to include one genome per genus. Downloaded genome phylum taxonomy was determined by cross-referencing the RefSeq genome ID with the GTDB-tk (R202 database) metadata.

**Reporting summary.** Further information on research design is available in the Nature Research Reporting Summary linked to this article.

## Data availability

The raw anaerobic digester sequencing data are available at the ENA with the bio project ID PRJEB48021, while the Zymo mock community raw sequencing data are available at PRJEB48692 (Supplementary Table 4). The UniProt TrEMBL database used in the study is available at https://ftp.uniprot.org/pub/databases/uniprot/previous_releases/release-2021_01/knowledgebase. The GTDB-tk database used in the study is available at https://data.ace.uq.edu.au/public/gtdb/data/releases/release202. Links for accessing the genome assemblies, MAGs and summary data

are available at https://github.com/Serka-M/Digester-MultiSequencing. Zymo Mock community reference sequences are available at https://s3.amazonaws.com/zymo-files/BioPool/D6322.refseq.zip. The NCBI RefSeq genome database is available at https://ftp.ncbi.nlm.nih.gov/genomes/refseq.

## Code availability

Links for accessing code used to generate figures as well as supplementary resources are available at https://github.com/Serka-M/Digester-MultiSequencing. Software tools used in the study are either referenced or are provided as links in the Methods section.

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

## Acknowledgements

The authors thank the plant operators at Fredericia Wastewater Treatment Plant for supplying the sample material. The study was funded by research grants from VILLUM FONDEN (15510) and the Poul Due Jensen Foundation (Microflora Danica).

## Author contributions

M.S. and R.H.K. performed DNA extraction and sequencing of the anaerobic digester, and selected the Zymo mock samples. R.D.W. prepared and sequenced the Zymo mock using R9.4.1 and Illumina. M.S., R.H.K. and M.A. wrote the first draft of the manuscript. S.M.K., T.Y.M., R.D.W. and E.A.S. contributed to experiment design, result interpretation and writing of the manuscript. All authors reviewed the manuscript.

## Competing interests

E.A.S., S.M.K., M.A., R.H.K. and R.D.W. are employed at DNASense ApS, which provides consulting and sequencing services. RHK, S.M.K. and T.Y.M. own shares in Oxford Nanopore Technologies PLC. The remaining author has no competing interests.

## Additional information

**Correspondence and requests for materials** should be addressed to Mads Albertsen.

# Reporting Summary

Nature Research wishes to improve the reproducibility of the work that we publish. This form provides structure for consistency and transparency in reporting. For further information on Nature Research policies, see our Editorial Policies and the Editorial Policy Checklist.

## Statistics

For all statistical analyses, confirm that the following items are present in the figure legend, table legend, main text, or Methods section.

| n/a | Confirmed | |
|---|---|---|
| ☐ | ☒ | The exact sample size ($n$) for each experimental group/condition, given as a discrete number and unit of measurement |
| ☒ | ☐ | A statement on whether measurements were taken from distinct samples or whether the same sample was measured repeatedly |
| ☒ | ☐ | The statistical test(s) used AND whether they are one- or two-sided<br>*Only common tests should be described solely by name; describe more complex techniques in the Methods section.* |
| ☒ | ☐ | A description of all covariates tested |
| ☒ | ☐ | A description of any assumptions or corrections, such as tests of normality and adjustment for multiple comparisons |
| ☐ | ☒ | A full description of the statistical parameters including central tendency (e.g. means) or other basic estimates (e.g. regression coefficient) AND variation (e.g. standard deviation) or associated estimates of uncertainty (e.g. confidence intervals) |
| ☒ | ☐ | For null hypothesis testing, the test statistic (e.g. $F$, $t$, $r$) with confidence intervals, effect sizes, degrees of freedom and $P$ value noted<br>*Give P values as exact values whenever suitable.* |
| ☒ | ☐ | For Bayesian analysis, information on the choice of priors and Markov chain Monte Carlo settings |
| ☒ | ☐ | For hierarchical and complex designs, identification of the appropriate level for tests and full reporting of outcomes |
| ☒ | ☐ | Estimates of effect sizes (e.g. Cohen's $d$, Pearson's $r$), indicating how they were calculated |

*Our web collection on statistics for biologists contains articles on many of the points above.*

## Software and code

Policy information about availability of computer code

| Data collection | MinKNOW software v21.05.25 (Oxford Nanopore, England) and Guppy v5.0.16 (Oxford Nanopore, England) |
|---|---|
| Data analysis | Cutadapt (v1.16), duplex-tools (v0.2.5, Oxford Nanopore), Porechop (v0.2.3), NanoFilt (v2.6.0), CCS (v6.0.0 Pacific Biosciences), NanoPlot (v1.24.0), Rasusa (v0.3.0), seqtk (v1.3), Counterr (v0.1), Flye (v2.9), Minimap2 (v2.17), Racon (v1.3.3), Medaka (v1.4.4, Oxford Nanopore), Megahit (v1.1.4), MetaBAT (v2.12.1), MaxBin2 (v2.2.7), Vamb (v3.0.2), DAS Tool (v1.1.2), CoverM (v0.6.1), CheckM (1.1.2), GTDB-tk (v1.5.0), tRNAscan-SE (v2.0.5), Prodigal (v2.6.3), Bowtie2 (v2.4.2), SAMtools (v1.9), CMseq (v1.0.3), dRep (v2.6.2), Diamond (v2.0.6), QUAST (v4.6.3), FastANI (v1.33), Barrnap (v0.9), Bioawk (v. 1.0), ncbi-genome-download (v0.3.0), infoseq (v. 6.6.0.0), https://github.com/Serka-M/Digester-MultiSequencing |

For manuscripts utilizing custom algorithms or software that are central to the research but not yet described in published literature, software must be made available to editors and reviewers. We strongly encourage code deposition in a community repository (e.g. GitHub). See the Nature Research guidelines for submitting code & software for further information.

## Data

Policy information about availability of data

All manuscripts must include a data availability statement. This statement should provide the following information, where applicable:
- Accession codes, unique identifiers, or web links for publicly available datasets
- A list of figures that have associated raw data
- A description of any restrictions on data availability

Raw sequencing reads are available on the European Nucleotide Archive: PRJEB48692 for the Zymo Mock microbial community and PRJEB48021 for the anaerobic digester sequencing data.
Bacterial genome assembly and additional files used in the study are available for download at https://doi.org/10.6084/m9.figshare.17008801.v1

# Field-specific reporting

Please select the one below that is the best fit for your research. If you are not sure, read the appropriate sections before making your selection.

☒ Life sciences ☐ Behavioural & social sciences ☐ Ecological, evolutionary & environmental sciences

For a reference copy of the document with all sections, see nature.com/documents/nr-reporting-summary-flat.pdf

# Life sciences study design

All studies must disclose on these points even when the disclosure is negative.

| | |
|---|---|
| Sample size | We carried out sequencing of the Zymo Mock microbial community using three different sequencing strategies (ONT R9.4.1, ONT R10.4, Illumina MiSeq) to compare performance. The Zymo Mock was chosen as the composition of the community is know and the reference sequences are publicly available. For each sequencing strategy the Zymo Mock was sequenced at a depth providing at least 100x coverage for the bacterial species of the Zymo mock, which was deemed as sufficient for downstream analysis.<br><br>The single anaerobic digester sample was chosen as a proxy for a complex microbial community and was sequenced using four different sequencing strategies (ONT R9.4.1, ONT R10.4, Illumina MiSeq, PacBio HiFi) to compare performance. For each sequencing strategy, the anaerobic digester sample was sequenced at a minimal sequencing depth of 12 Gb, which, from previous projects, was expected to provide a sufficient amount of metagenome assembled genomes for downstream analysis. |
| Data exclusions | For comparing genome bins, bins which did not cluster between the different sequencing approaches, were excluded from direct comparisons. Also, long-read-based bins, which featured lower Illumina read coverage than 5, were excluded from direct comparisons. For the IDEEL test, query matches, which were not present in all datasets, were omitted to reduce noise. Also for the IDEEL test, the R.9.4.1 read dataset was sub-sampled to acquire bins at comparable coverage levels to other sequencing strategies. |
| Replication | For the Zymo Mock community, sequencing was performed independently. Zymo Mock bacterial genome assemblies were generated at multiple coverage levels to assess the impact of sequencing depth but also to assess the variability in genome quality metrics once adequate sequencing depth was achieved.<br><br>The anaerobic digester sludge sample was sequenced using 3 different sequencing methods (Illumina, PacBio CCS, Nanopore) and 2 different Nanopore chemistries (R9.4.1 and R10.4). Hence, the DNA sample from the anaerobic digester was sequenced 5 times (independently), but no biological replicates have been included in the study.<br><br>Sequencing of the 9 additional time-series Illumina datasets (no technical replicates) of the same anaerobic digester was performed independently from this study. |
| Randomization | Not relevant since this project does not use experimental groups |
| Blinding | Not relevant since this project does not use experimental groups |

# Reporting for specific materials, systems and methods

We require information from authors about some types of materials, experimental systems and methods used in many studies. Here, indicate whether each material, system or method listed is relevant to your study. If you are not sure if a list item applies to your research, read the appropriate section before selecting a response.

### Materials & experimental systems

| n/a | Involved in the study |
|---|---|
| ☒ | Antibodies |
| ☒ | Eukaryotic cell lines |
| ☒ | Palaeontology and archaeology |
| ☒ | Animals and other organisms |
| ☒ | Human research participants |
| ☒ | Clinical data |
| ☒ | Dual use research of concern |

### Methods

| n/a | Involved in the study |
|---|---|
| ☒ | ChIP-seq |
| ☒ | Flow cytometry |
| ☒ | MRI-based neuroimaging |

