## [Peer Review File · Nature Methods]

Peer Review Information

Manuscript Title: Oxford Nanopore R10.4 long-read sequencing enables near-finished bacterial genomes from pure cultures and metagenomes without short-read or reference polishing

Corresponding author name: Mads Albertsen

Reviewer Comments & Decisions:

Decision Letter, initial version:

Subject: Decision on Nature Methods submission NMETH-BC47580
Message:

15th Feb 2022

Dear Dr Albertsen,

We apologize again for the longer peer review of your Brief Communication, "Oxford Nanopore R10.4 long-read sequencing enables near-perfect bacterial genomes from pure cultures and metagenomes without short-read or reference polishing" than we would hope. This manuscript has now been seen by 2 reviewers. As you will see from their comments below, although the reviewers find your work of potential interest, they have raised a number of concerns. We are interested in the possibility of publishing your paper in Nature Methods, but would like to consider your response to these concerns before we reach a final decision on publication.

We therefore invite you to revise your manuscript to address all the concerns raised by the two reviewers.

- * include a point-by-point response to the reviewers and to any editorial suggestions
- * please underline/highlight any additions to the text or areas with other significant changes to facilitate review of the revised manuscript
- * address the points listed described below to conform to our open science requirements

2* ensure it complies with our general format requirements as set out in our guide to authors at www.nature.com/naturemethods

* resubmit all the necessary files electronically by using the link below to access your home page

[REDACTED]

We hope to receive your revised paper within eight weeks. We are very aware of the difficulties caused by the COVID-19 pandemic to the community. If you cannot send it within this time, please let us know. In this event, we will still be happy to reconsider your paper at a later date so long as nothing similar has been accepted for publication at Nature Methods or published elsewhere.

OPEN SCIENCE REQUIREMENTS

REPORTING SUMMARY AND EDITORIAL POLICY CHECKLISTS

Please note that these forms are dynamic 'smart pdfs' and must therefore be downloaded and completed in Adobe Reader. We will then flatten them for ease of use by the reviewers. If you would

like to reference the guidance text as you complete the template, please access these flattened versions at <http://www.nature.com/authors/policies/availability.html>.

DATA AVAILABILITY

We strongly encourage you to deposit all new data associated with the paper in a persistent repository where they can be freely and enduringly accessed. We recommend submitting the data to discipline-specific and community-recognized repositories; a list of repositories is provided here:

<http://www.nature.com/sdata/policies/repositories>

All novel DNA and RNA sequencing data, protein sequences, genetic polymorphisms, linked genotype and phenotype data, gene expression data, macromolecular structures, and proteomics data must be deposited in a publicly accessible database, and accession codes and associated hyperlinks must be provided in the “Data Availability” section.

Please include a “Data availability” subsection in the Online Methods. This section should inform readers about the availability of the data used to support the conclusions of your study, including accession codes to public repositories, references to source data that may be published alongside the paper, unique identifiers such as URLs to data repository entries, or data set DOIs, and any other statement about data availability. At a minimum, you should include the following statement: “The data that support the findings of this study are available from the corresponding author upon request”, describing which data is available upon request and mentioning any restrictions on availability. If DOIs are provided, please include these in the Reference list (authors, title, publisher (repository name),

identifier, year). For more guidance on how to write this section please see:

<http://www.nature.com/authors/policies/data/data-availability-statements-data-citations.pdf>

CODE AVAILABILITY

Please include a “Code Availability” subsection in the Online Methods which details how your custom code is made available. Only in rare cases (where code is not central to the main conclusions of the paper) is the statement “available upon request” allowed (and reasons should be specified).

MATERIALS AVAILABILITY

ORCID

Nature Methods is committed to improving transparency in authorship. As part of our efforts in this direction, we are now requesting that all authors identified as ‘corresponding author’ on published papers create and link their Open Researcher and Contributor Identifier (ORCID) with their account on the Manuscript Tracking System (MTS), prior to acceptance. This applies to primary research papers only. ORCID helps the scientific community achieve unambiguous attribution of all scholarly contributions. You can create and link your ORCID from the home page of the MTS by clicking on

'Modify my Springer Nature account'. For more information please visit www.springernature.com/orcid.

Sincerely,

Lin Tang, PhD
Senior Editor
Nature Methods

Reviewers' Comments:

Reviewer #1:

Remarks to the Author:

The authors present evidence that the latest Oxford Nanopore sequencing chemistry (R10.4) no longer needs short read polishing to achieve highly accurate genomes. This is a reasonably significant report if true, and represents a step change for the technology. It is a paper of general interest, overturns the current consensus opinion that ONT data contains many errors, and is of potential interest to Nature Methods.

There are a few things that would improve the manuscript, and provide better/more evidence about the authors' claims

The term "near-perfect" is used several times, but this is not well defined. Could the authors perhaps suggest a definition for a "near perfect" genome?

A few relevant papers are perhaps missing from the introduction:

5- first single contig nanopore assemblies from metagenomes: <https://www.nature.com/articles/s41587-020-0422-6>, <https://www.nature.com/articles/s41587-019-0202-3>

- first single contig hybrid nanopore and illumina genome:
<https://gigascience.biomedcentral.com/articles/10.1186/s13742-015-0101-6>

Should the Zymo sample not also be sequenced using PacBio HiFi?

The authors appear to use HiFi and CCS interchangeably - is this deliberate?

Why is the fungus not included in Figure 1?

The IDEEL score for *S. enterica* is far lower than for other species but this is not mentioned anywhere

The major take home message for figure 1 appears to be that R10.4 has higher quality and is better at homopolymers than R9.4, and as a result, R10.4 is as good as R9.4+Illumina. If this is the authors' message, then it should be clearly stated. Also, given 40X appears to be a cutoff the authors are proposing, this could be highlighted in the plots

The authors mention that comparisons between technologies are difficult because of different yields and read lengths. We have encountered this issue before, and solved it by using a script called `common_length_distributions.sh` from this repo: <https://github.com/makovalab-psu/NoiseCancellingRepeatFinder/tree/master/experiments>. This script can take two FASTA files and subsample them so they have the same sequence length characteristics and yield. Once this has been done, a platform comparison is possible. The authors may wish to do this.

In table 1, R10.4 results are worse than R9.4 in most cases, this is probably due to yield, and here I think sub-sampling would be useful

It is not clear to me whether Illumina correction is applied to the raw metagenome assembly or to the MAGs. I would have thought it better to correct the raw metagenome assembly using Illumina as (i) this will get better unbiased mapping and (ii) the corrected assembly should bin better. However, I get the impression correction was applied to the MAGs. Could the authors justify this?

In Figure 2, I think the PacBio CCS data should also be Illumina corrected, for completeness. The major result here is that for coverage > 40X, R10.4 doesn't benefit much from correction whereas R9.4 does.

6However, there is an outlier in the R10.4 data and this should be discussed. Also, the 40X cut-off should be highlighted in the plots. The result is also biased as there are simply a lot more genomes at greater than 40X in the R9.4 data than in the R10.4 data, which potentially biases the results.

For data availability, I would like to see:

- the basecalled FASTQ from the nanopore data submitted (i.e. not just FAST5)
- raw genome and metagenome assemblies submitted
- All MAGs submitted, alongside completeness and contamination

When the authors describe size selection for nanopore or PacBio, they should provide more details (i.e. traces, and what size range was selected)

I note that MinION, PromethION and GridION were all used in this study. These platforms do differ in their data quality and this should be discussed.

Reviewer #2:

Remarks to the Author:

The paper of Sereika et al builds on earlier work of this group, and the work of other groups, on the recovery procedures for long read metagenome-assembled genomes. In the present paper, the authors show the advantages of ONT 10.4 sequencing for obtaining improved MAG sequence quality, particularly in relation to the accurate capture of homopolymer runs. The findings and the data are certainly useful to workers in this field. However, I have a number of issues about the way these data are being presented, as well as some of the conceptual underpinnings, in the present manuscript. I have included some comments and suggestions below.

Definitions of “near perfect genomes”: the authors make some broad claims about the ability of ONT 10.4 sequencing to generate improved genome sequence compared to ONT 9.4 sequencing, however there is no formal definition provided of the terms “perfect” or “near perfect”. This is a critical omission, because there is no way for the reader to assess the data or the claims. Conceptually, at least, evoking the term “perfect” is highly problematic for both methodological reasons in general and ecological reasons in particular.

For example, line 43 “...near-perfect microbial reference [Zymo mock data] genomes can be obtained from R10.4 data alone”: not all the relevant data has been included to support this claim, notably the lack of assembly statistics. A reasonable starting point for a near perfect (bacterial) genome would be one that assembled into a single chromosomal sequence, either closed or otherwise (assuming it is not actually multipartite genome). Is this actually the case or not? Can the authors provide the assembly statistics for this analysis? If this is not the case, how do the authors regard their claims? For example, if a genome does not assemble into a single chromosome, and comprised of, say, 4 contigs, each of which are comprised of high quality genic and non-genic sequence, would the authors consider this to be a near perfect genome or not?

Homopolymer analysis for Zymo mock data.

“improved ability to call homopolymers, as R10.4 is able to correctly call 47 the length of the majority of homopolymers up to a length of 10”. I agree 10.4 is an improvement on 9.4, but this is somewhat overstated. See Fig 1B and Fig S3 (the distributions for the C-runs on 10.4 are actually rather flat, for runs above 6).

IDEEL. Given that IDEEL statistics can be well above unity (as well as obtain spuriously high values due to secondary alignments), it would be useful if some (IQR and range) was presented, so the reader can get a better sense of their accuracy and precision.

Table 1 and Fig S6:

I am lumping these together because there seem to be a number of ambiguities or seeming inconsistencies, so I would appreciate if they could be explained clearly and accurately.

Firstly, in the Methods, it is actually quite hard to figure out what data has been incorporated in the binning procedures: can you simply list each of the analyses that are being included? (so the reader doesn't have to work so hard!). I presume that you are using the Illumina data to guide the binning, but that you are only considering the long read component of the bins in your analysis? Is that correct or not? In terms of the writing, it is also difficult to distinguish when you are referring to the use of Illumina data for binning and when it is being used for sequence error correction.

In Table 1 what does the “/+” mean in the second and third columns? Does this mean that in these results are from binning on the combined long read and short read derived contigs? What does “Contigs pr. HQ MAG (median)” actually mean? And why are there two numbers here? And are we meant to interpret them? Some clear, unambiguous table notes would be extremely useful here.

Table 1 has some key information missing, that is only discernable in Fig S6. For example, it appears as if the mean number of contigs per bin, from either PacBio or ONT, is around around 15 (I would guess based on the y-axis). This is critical information that should be included in Table 1, as it speaks directly to the overall recoveabilty of genomes.

In Table 1, why are the results for ONT-only not reported?

In Table 1, how many of these HQ-MAGs are single chromosomal sequences, whether circular otherwise?

In Fig S6A, I could not find results relating to combination of ONT+Illumina data. Given these seem to be reported in Table 1 [but see my comment above], why are they omitted here?

Fig S6B. Bulk genome quality statistics, why is contamination not being shown here? Does strain heterogeneity correlate with the polymorphic rate statistics?

(line 17) "...can be used to generate near-perfect microbial genomes from isolates or metagenomes without short-read or reference polishing". While that is technically correct, in the context of the real community analysis, one clearly still needs short read data to obtain the genomes in the first place (Table 1). It seems somewhat disingenuous to claim that for the consideration of local sequence quality, whether coding or non-coding, 10.4 sequencing obviates the need for Illumina sequencing, while ignoring the limitation that the underlying genomes would not, in all likelihood, be recovered without the use of the same data.

Author Rebuttal to Initial comments

Reviewer #1:

Remarks to the Author:

The authors present evidence that the latest Oxford Nanopore sequencing chemistry (R10.4) no longer needs short read polishing to achieve highly accurate genomes. This is a reasonably significant report if true, and represents a step change for the technology. It is a paper of general interest, overturns the current consensus opinion that ONT data contains many errors, and is of potential interest to Nature Methods.

There are a few things that would improve the manuscript, and provide better/more evidence about the authors' claims

9Q1.1.1: The term "near-perfect" is used several times, but this is not well defined. Could the authors perhaps suggest a definition for a "near perfect" genome?

A1.1.1 comment: This question was raised by both reviewers (Q1.2.1), and we agree that we did not define it properly in the manuscript. It's a complex discussion and we have tried to summarize our reasoning below.

In 2017, numerous groups defined quality standards (MIMAG) for MAGs (Bowers et al., 2017). They defined "finished" genomes as "Single contiguous sequence without gaps or ambiguities with a consensus error rate equivalent to Q50 or better". Recently, Wick et al. 2021 has demonstrated that errors in genome consensus sequences can be (are) introduced by assemblers and polishing tools, making the quality cutoff of Q50 for a "finished" genome to be unrealistic in many scenarios.

One of the main issues with the currently defined standards are that bacterial genomes can meet the second tier of genome quality "High quality", while still featuring a high rate of genome consensus errors and protein truncations when based on long-read sequencing (especially Nanopore), as nicely demonstrated by Watson & Warr in 2019. This has led to the need for short-read polishing of bacterial genome data when using Nanopore sequencing to obtain "truly" high-quality / finished genomes. The "high-quality" definition has also received critique from Chen et al., 2020.

Here, we showcase that Nanopore R10.4 can be used, without short read polishing, to recover bacterial genomes, which feature a much greater consensus quality than that of the commonly used "high quality" genomes (and MAGs). Hence, we feel that we needed a term that underscored this significant milestone in Nanopore sequencing.

We were careful not to use the word "finished" as there can be few systematic errors left in genomes made using Nanopore R10.4 data (see later theoretical analysis of homopolymers in all RefSeq genomes) and as "finished" also requires a single contig assembly.

This was our main reason for using the term "near-perfect" as we seem to be very close to the "perfect" genome as also discussed by Wick et al., 2021. In hindsight, it would have been more appropriate to use "near-finished" to stay closer to the MIMAG definitions laid out by Bowers et al., 2017 and we suggest to use this term in the updated manuscript.

We do not state that Nanopore R10.4 sequencing always results in "near-finished" quality genomes, but that it is now possible to obtain near-finished genomes using Nanopore R10.4 alone for the vast majority of bacterial genomes. To further substantiate this claim we analysed the homopolymer content in all complete genomes (1 pr. genera) in the NCBI RefSeq database. We found that only 1% of genomes had more than 1 homopolymer (>10 bp) pr. 100.000 bp (Q50). Hence, given that R10.4 can accurately resolve

10most homopolymers of up to 11 bp, homopolymers should have very little impact on consensus quality for the vast majority of bacterial genomes. This theoretical finding is very similar to our real-life observation in both pure cultures and metagenomes, where short-read polishing does not significantly improve consensus quality at a coverage of >40x Nanopore R10.4 data.

Hence, we would define a “near-finished” bacterial genome (or MAG), as a high-quality genome, which does not significantly improve in consensus quality from short read polishing.

Bowers, R., Kyrpides, N., Stepanauskas, R. et al. Minimum information about a single amplified genome (MISAG) and a metagenome-assembled genome (MIMAG) of bacteria and archaea. *Nat Biotechnol* 35, 725–731 (2017). <https://doi.org/10.1038/nbt.3893>

Watson, M., Warr, A. Errors in long-read assemblies can critically affect protein prediction. *Nat Biotechnol* 37, 124–126 (2019). <https://doi.org/10.1038/s41587-018-0004-z>

Chen et al. Accurate and complete genomes from metagenomes. *Genome Research* 30, 315-333 (2020). Doi: 10.1101/gr.258640.119

Wick, R.R., Judd, L.M., Cerdeira, L.T. et al. Tricycler: consensus long-read assemblies for bacterial genomes. *Genome Biol* 22, 266 (2021). <https://doi.org/10.1186/s13059-021-02483-z>

A1.1.1 changes made: We have changed the term “near-perfect” to “near-finished” and provided an explanation in the revised manuscript. Furthermore, we have analysed the homopolymer content in all available complete RefSeq genomes to demonstrate that large homopolymers are rare in bacteria and added it as **Figure S8** along with a discussion of the analysis in the manuscript.

Q1.1.2: A few relevant papers are perhaps missing from the introduction:

- first single contig nanopore assemblies from metagenomes: <https://www.nature.com/articles/s41587-020-0422-6>, <https://www.nature.com/articles/s41587-019-0202-3>

- first single contig hybrid nanopore and illumina genome: <https://gigascience.biomedcentral.com/articles/10.1186/s13742-015-0101-6>

A1.1.2 comment: We would like to thank the reviewer for recommending highly relevant literature.

A1.1.2 changes made: The recommended articles have been cited in the revised introduction.

Q1.1.3: Should the Zymo sample not also be sequenced using PacBio HiFi?

A1.1.3 comment: PacBio HiFi sequencing of the Zymo sample was not performed, since genome reference sequences were available from Zymo Research. The main goal of the Zymo sample analysis is to assess the performance of different Nanopore sequencing chemistries (with and without short read polishing) at reconstructing the consensus sequences of the Zymo Mock community. From short-read based SNP analysis some of the reference genomes either has assembly problems or true biological divergence from the Zymo Lots we sequenced (see **Table S3**). We could have used the Illumina short-read data to polish the Zymo reference sequences. However, we choose not to as it would complicate other researchers efforts to reproduce our results and would not impact our analysis as we are only interested in relative improvements (unpolished vs. polished), not absolute values.

For the digester sample, PacBio HiFi sequencing was performed as reference genomes sequences were unavailable and these were needed to make direct quality comparisons between the different Nanopore chemistries. PacBio HiFi data is the current gold-standard in the field and was used to generate as good reference genomes as possible (although relatively expensive). Based on the excellent suggestion by the reviewer (**Q1.1.11**) we have now also polished the PacBio HiFi assembly using short reads which improves the consensus sequence of low-coverage PacBio bins.

A1.1.3 changes made: A note was added to the Materials and methods section that Zymo reference sequences were used as a substitute to PacBio HiFi. Furthermore, the PacBio HiFi assembly is now also polished with Illumina data to obtain the best-possible reference data.

Q1.1.4: The authors appear to use HiFi and CCS interchangeably - is this deliberate?

A1.1.4 comment: We apologize for the confusing use of PacBio CCS and HiFi. The two terms refer to the same sequencing technology, but we should be consistent in the usage.

A1.1.4 changes made: We have altered the naming to only use “PacBio HiFi” to avoid potential confusion.

Q1.1.5: Why is the fungus not included in Figure 1?

A1.1.5 comment: Initially, we intended to include the *Saccharomyces Cerevisiae* genome statistics in the comparison. However, the fungal genome is scarce in the Zymo Mock sample, featuring an approximate coverage of 35 in the R10.4 dataset, a coverage of 27 in the R9.4.1 data, and a coverage of about 10 in the Illumina dataset. Hence, the *Saccharomyces Cerevisiae* genome was excluded from the Zymo Mock sample analysis due to insufficient coverage by the sequencing datasets.

Also, even with enough coverage, we do not wish to make conclusions about the ability of Nanopore R10.4 to obtain high-quality fungal genomes given the more complex structure of these.

A1.1.5 changes made: A note was added to the Materials and methods section about the exclusion of *Saccharomyces Cerevisiae* genome from the analysis.

Q1.1.6: The IDEEL score for *S enterica* is far lower than for other species but this is not mentioned anywhere

A1.1.6 comment: The IDEEL test is often being viewed as a reference-free approach for assessing frameshift errors. However, the method relies on databases and if only distantly related organisms exists in the database, this will tend to decrease the IDEEL score. In general, a variation of 1-5 IDEEL score points between different genomes can be expected as part of standard deviation and has been reported in a previous study (Wick et al., 2021).

However, the focus of applying the IDEEL test in this study was to measure the difference in IDEEL scores within the same genome before and after applying Illumina short read polishing. Hence, we use the IDEEL score as a relative measurement, which thereby is not as dependent on the reference database.

Wick, R.R., Judd, L.M., Cerdeira, L.T. et al. Tricycler: consensus long-read assemblies for bacterial genomes. *Genome Biol* 22, 266 (2021). <https://doi.org/10.1186/s13059-021-02483-z>

A1.1.6 changes made: A note on the IDEEL score was added to provide further clarification on why the relative improvement is the most relevant metric in our use-case.

Q1.1.7: The major take home message for figure 1 appears to be that R10.4 has higher quality and is better at homopolymers than R9.4, and as a result, R10.4 is as good as R9.4+Illumina. If this is the authors' message, then it should be clearly stated. Also, given 40X appears to be a cutoff the authors are proposing, this could be highlighted in the plots

A1.1.7 comment: The message that we are trying to convey is that bacterial genomes or MAGs acquired from Nanopore-only data (R10.4 chemistry, 40x coverage) no longer feature significant amounts of systematic errors and protein truncations, which is a major bottleneck for Nanopore R9.4.1 (and earlier) sequencing chemistries. Hence, R10.4 is equivalent to R9.4 + Illumina as the reviewer states.

A1.1.7 changes made: Edits to the main text have been made to provide further clarification. A dashed line at 40x coverage was added to all relevant plots.

Q1.1.8: The authors mention that comparisons between technologies are difficult because of different yields and read lengths. We have encountered this issue before, and solved it by using a script called `common_length_distributions.sh` from this repo: <https://github.com/makovalab-psu/NoiseCancellingRepeatFinder/tree/master/experiments>. This script can take two FASTA files and subsample them so they have the same sequence length characteristics and yield. Once this has been done, a platform comparison is possible. The authors may wish to do this.

A1.1.8 comment: We would like to thank the reviewer for the recommendation. However, the read length distributions between PacBio and Nanopore datasets are such that picking only overlapping values would result in a miniscule amount of data for all platforms (see **Figure S4**), significantly hindering MAG recovery and thus compromising analysis results. Furthermore, even with the exact same length distribution, comparisons would not be easy due to large differences in error-profiles between the different chemistries.

A1.1.8 changes made: We thank the reviewer to get us to think about how to best make the datasets comparable. Hence, for the IDEEL test with complex metagenome data, we have subsampled the R9.4.1 dataset (note added to the Material and methods) to generate MAGs at comparable coverage levels to R.10.4 and PacBio datasets. We feel that this greatly enhances the direct visual comparison of the impact of coverage on the data quality (see the updated **Figure 2**).

Q1.1.9: In table 1, R10.4 results are worse than R9.4 in most cases, this is probably due to yield, and here I think sub-sampling would be useful

A1.1.9 comment: We agree with the reviewer that higher coverage leads to more contiguous assemblies. One additional point is that, currently, R10.4 features a significant rate of concatemers (multiple DNA sequences joined into a single read). While we bioinformatically split most of the concatemers, we still assume that some concatemers bypass the splitting process and make it to the assembly, resulting in an additionally fragmented metagenome. We expect the concatemer read problem to be temporary and be fully solved via future software improvements, hence we have not described this in the main text and just mentioned it in the methods section.

A1.1.9 changes made: We have subsampled the R9.4.1 dataset (see **A1.1.8 changes made**) and added a note to the Materials and methods section about concatemers in R10.4 data.

Q1.1.10: It is not clear to me whether Illumina correction is applied to the raw metagenome assembly or to the MAGs. I would have thought it better to correct the raw metagenome assembly using Illumina as (i) this will get better unbiased mapping and (ii) the corrected assembly should bin better. However, I get the impression correction was applied to the MAGs. Could the authors justify this?

14A1.1.10 comment: We thank the reviewer for highlighting that we have not explained our methods clearly. Illumina short read correction was applied to the assembled metagenome before performing the binning, exactly due to the potential problems mentioned by the reviewer.

A1.1.10 changes made: An edit to the “Read assembly and binning” part of the Materials and methods section was made to provide further clarification. We have also added a schematic, describing the main bioinformatics processes with complex metagenome data, to the supplementary material (**Figure S9**).

Q1.1.11: In Figure 2, I think the PacBio CCS data should also be Illumina corrected, for completeness. The major result here is that for coverage > 40X, R10.4 doesn't benefit much from correction whereas R9.4 does. However, there is an outlier in the R10.4 data and this should be discussed. Also, the 40X cut-off should be highlighted in the plots. The result is also biased as there are simply a lot more genomes at greater than 40X in the R9.4 data than in the R10.4 data, which potentially biases the results.

A1.1.11 comment: We thank the reviewer for the excellent suggestion and have updated the IDEEL plot to include Illumina-corrected PacBio bins, as well as results for the subsampled R9.4 dataset. We would also like to point out the outlier mentioned by the reviewer is actually present in all datasets and features a similar IDEEL score across all long-read platforms.

The outlier MAG in question gets classified as Patescibacteria in all datasets and no close relatives are present in the database, hence we suspect distant database entries to be the cause of the systematically lower IDEEL score. The other outlier in the revised IDEEL plot is for the R9.4.1 MAG at > 100x coverage, featuring an IDEEL score improvement of 71 from short read polishing. Homopolymers were counted in the Illumina-polished version of the MAG to find 594 homopolymers of length 8 or higher. Hence, the exceptionally high long homopolymer counts are expected to be the cause for the low IDEEL score of the R9.4.1-only MAG.

A1.1.11 changes made: The IDEEL test for the complex metagenome data was repeated with subsampled R.9.4 read data as well as with Illumina-polished PacBio metagenome data. A 40x coverage cut-off is highlighted in the plot. Notes on the outliers were added to the manuscript. Also, MAGs were re-clustered with only long read datasets (Illumina-only MAGs were not used), which allowed to include more MAGs in the IDEEL plot (**Figure 2**).

Q1.1.12: For data availability, I would like to see:

- the basecalled FASTQ from the nanopore data submitted (i.e. not just FAST5)
- raw genome and metagenome assemblies submitted

15- All MAGs submitted, alongside completeness and contamination

A1.1.12 comment: All data the reviewer mention was already available, but we thank the reviewer for highlighting that it was not obvious from the manuscript as much was stated in the methods section. The raw Illumina, Nanopore and PacBio data is available at ENA. Metagenome and Zymo sample assemblies are available via Figshare (<https://doi.org/10.6084/m9.figshare.17008801.v1>). Dataframes, which include MAG completeness and contamination values, have been deposited at a GitHub repository (<https://github.com/Serka-M/Digester-MultiSequencing>). The github repository also displays all analysis in R markdown as well as coordinating the addition of new analysis/dataframes.

We have not submitted the assemblies and MAGs to ENA directly as we feel that it would inflate the databases with numerous almost identical versions of the same MAGs.

A1.1.12 changes made: An edit to the “Data availability” section was made to provide further clarification.

Q1.1.13: When the authors describe size selection for nanopore or PacBio, they should provide more details (i.e. traces, and what size range was selected)

A1.1.13 comment: We applied DNA size selection to deplete DNA fragments below 10 kb length using the Circulomics SRE XS kit, and the sequencing service provider, which performed PacBio sequencing, additionally size selected the sample using Blue Pippin to select for DNA fragments above 10 kb.

A1.1.13 changes made: Additional clarification in the Materials and Methods section was added about size selection of the DNA sample.

Q1.1.14: I note that MinION, PromethION and GridION were all used in this study. These platforms do differ in their data quality and this should be discussed.

A1.1.14 comment: While the MinION and the GridION are different devices, we would like to note that both use the same type of Nanopore flow cells and basecalling is performed using the same Guppy models. Data output from the GridION/MinION is comparable in terms of quality and, if low quality data is generated, it is caused by either poor sequencing library preparation (e.g. loading the flowcell with contaminants because of insufficient purification) or inadequate sequencing run conditions (e.g. the sequencer is in an environment with either too high or too low temperature), but not the difference between the two devices.

The PromethION uses a different model of flowcells, which contains more pores allowing for higher sequencing yields to be generated, although the underlying sequencing chemistry (the nanopore itself) is the same as in the GridION/MinION. Hence, the essential factor for Nanopore data quality is not the device of choice, but the sequencing chemistry (eg. R10.4 vs R9.4.1) and the systematic errors associated with it. In other words, we expect the systematic errors for R9.4.1 chemistry to be present in the data regardless of whether the read data was generated on a GridION or a PromethION.

In our experience, variations in read quality between different Nanopore sequencing runs can be observed, although they are usually attributed to technical variation, expertise of the person performing the sequencing experiment. Slight variation in read data quality between different GridION/PromethION runs was also reported by Nicholls et al. 2019.

In our study, Nanopore reads have been quality filtered to remove low quality reads generated during sequencing, thus making the Nanopore data more comparable.

Samuel M Nicholls, Joshua C Quick, Shuiquan Tang, Nicholas J Loman, Ultra-deep, long-read nanopore sequencing of mock microbial community standards, GigaScience, Volume 8, Issue 5, May 2019, giz043, <https://doi.org/10.1093/gigascience/giz043>

A1.1.14 changes made: A note was added that slight variation in data quality between the sequencing runs is expected.

Reviewer #2:

Remarks to the Author:

The paper of Sereika et al builds on earlier of this group, and the work of other groups, on the recovery procedures for long read metagenome-assembled genomes. In the present paper, the authors show the advantages of ONT 10.4 sequencing for obtaining improved MAG sequence quality, particularly in relation to the accurate capture of homopolymer runs. The findings and the data are certainly useful to workers in this field. However, I have a number of issues about the way these data are being presented, as well as some of the conceptual underpinnings, in the present manuscript. I have included some comments and suggestions below.

Q1.2.1: Definitions of “near perfect genomes”: the authors make some broad claims about the ability of ONT 10.4 sequencing to generate improved genome sequence compared to ONT 9.4 sequencing, however there is no formal definition provided of the terms “perfect” or “near perfect”. This is a critical omission, because there is no way for the reader to assess the data or the claims. Conceptually, at least, evoking the term “perfect” is highly problematic for both methodological reasons in general and ecological reasons in particular.

A1.2.1 comment: We agree that this was insufficiently described in the manuscript and have commented on it in **A1.1.1 comment** to reviewer 1 who raised similar concerns.

A1.2.1 changes made: The relevant changes made are described in **A1.1.1 changes made**.

Q1.2.2: For example, line 43 “...near-perfect microbial reference [Zymo mock data] genomes can be obtained from R10.4 data alone”: not all the relevant data has been included to support this claim, notably the lack of assembly statistics. A reasonable starting point for a near perfect (bacterial) genome would be one that assembled into a single chromosomal sequence, either closed or otherwise (assuming it is not actually multipartite genome). Is this actually the case or not? Can the authors provide the assembly statistics for this analysis? If this is not the case, how do the authors regard their claims? For example, if a genome does not assemble into a single chromosome, and comprised of, say, 4 contigs, each of which are comprised of high quality genic and non-genic sequence, would the authors consider this to be a near perfect genome or not?

A1.2.2 comment: The assembly statistics for the Zymo mock species were acquired using QUAST and the dataframes are publically available (<https://github.com/Serka-M/Digester-MultiSequencing/blob/main/code/plotting-zymo/quast>). We did hundreds of comparisons and hence decided to make all data available in public data frames instead of extremely large supplementary tables. We apologize that this was not made clear in the manuscript.

At lower coverages (5-10) some assemblies are fragmented simply due to the lack of overlaps between reads. However, the fragmented sequences can still be aligned to the reference genomes to acquire consensus sequence quality metrics or predict protein sequences. At higher coverages, most Zymo species assemble into circular genomes (and a possible additional plasmid sequences).

We would like to state that our focus is on genome consensus quality, not the contiguity of the assembly as this is more directly tied to read lengths (and coverage). Other articles have extensively described how single contig assemblies are routinely achieved by long-read sequencing and that most bacterial genomes can be obtained in single contigs with read lengths above 7 kbp (see e.g. Koren and Phillippy 2015). Our paper describe that Nanopore R10.4 can now be used as a stand-alone technology, which we think is key development that marks a new era of democratized genome sequencing at unprecedented scale.

Koren S & Phillippy AM. One chromosome, one contig: complete microbial genomes from long-read sequencing and assembly. *Current Opinion in Microbiology*, 2015, 23:110-120.

A1.2.2 changes made: We have provided Zymo assembly statistics at 40x coverage as **Table S4** and made it more clear in the “data availability” section that all data is available online in the github

project repository.

Q1.2.3: Homopolymer analysis for Zymo mock data.

“improved ability to call homopolymers, as R10.4 is able to correctly call the length of the majority of homopolymers up to a length of 10”. I agree 10.4 is an improvement on 9.4, but this is somewhat overstated. See Fig 1B and Fig S3 (the distributions for the C-runs on 10.4 are actually rather flat, for runs above 6).

A1.2.3 comment: We agree with the reviewer that the basecaller is calling homopolymers of some nucleotides better than others in Nanopore R10.4 data. However, we would like to note that for R10.4 chemistry, the most commonly called cytosine homopolymers (while true length is 9) is still 9, hence we would still expect most of the cytosine homopolymers of length 9 to be called correctly in the genome consensus sequence via Medaka polishing when sufficient coverage is present (as showcased in Fig S7). Furthermore, when using Medaka for consensus polishing, both strands are used, hence homopolymer G's are also used to correct homopolymer C's.

To further provide evidence that calling homopolymers up to the length of 10 in bacterial genomes is a remarkably milestone to reach, we downloaded all bacterial and archaeal genomes from the NCBI RefSeq database (assembly level: complete) and counted homopolymers. In bacterial and archaeal genomes, homopolymer counts tend to decrease logarithmically as the length increases. Only 1% of genomes had a long homopolymer (>10) rate of more than 1 in 100,000 bp (theoretical Q50). Hence, for the vast majority of bacterial genomes (99% in the RefSeq database) we would expect that homopolymer related errors constitute less than 1 in 100,000 bp using Nanopore R10.4 data alone (Q50). This is a very important milestone to reach and why we are confident to call that near-finished genomes are now possible from Nanopore R10.4 data alone, provided a coverage of 40 x.

A1.2.3 changes made: We added a note on homopolymer calling accuracy for cytosines in the main text. We have also included our analysis of homopolymers in complete genomes in the NCBI RefSeq database as **Figure S8** and provide an description of the analysis in the main text. We thank the reviewer for this comment that prompted us to do a proper evaluation of the entire RefSeq database for homopolymer content.

Q1.2.4: IDEEL. Given that IDEEL statistics can be well above unity (as well as obtain spuriously high values due to secondary alignments), it would be useful if some (IQR and range) was presented, so the reader can get a better sense of their accuracy and precision.

A1.2.4 comment: As discussed in **A1.1.6**, the IDEEL test features some caveats, but is fairly robust

19when assessing the impact of short read polishing on correcting frameshift errors on a per-genome (or MAG) level. Hence, using the same approach as Wick et al., 2021, we consider the change in the IDEEL score for an **individual genome** before and after short read correction to be the main indicator of whether supplemental short read polishing was beneficial or not. Note that we used the “IDEEL score” and not individual IDEEL values in order to be able to compare large number of assemblies against each other in the same plot.

Wick, R.R., Judd, L.M., Cerdeira, L.T. et al. Tricycler: consensus long-read assemblies for bacterial genomes. *Genome Biol* 22, 266 (2021). <https://doi.org/10.1186/s13059-021-02483-z>

A1.2.4 changes made: Additional information on the IDEEL test and the IDEEL score was added to the main text as well as the Material and methods section to provide further clarification.

Q1.2.5: Table 1 and Fig S6: I am lumping these together because there seem are a number of ambiguities or seeming inconsistencies, so I would appreciate if they could be explained clearly and accurately.

A1.2.5 comment: We thank the reviewer for pointing this out. Table 1 and Figure S6 could definitely have been explained more clearly. Table 1 presents overall sequencing, assembly and binning results for the different sequencing platforms, whereas Fig S6 focuses on a subset of the data. This is because in order to directly compare MAG completeness, fragmentation or mismatch/indel rates between different sequencing platforms, we chose to only use MAGs that are observed in all datasets. This way, we can get per-MAG metrics across the different sequencing platforms (with and without short read polishing). We agree that this approach reduces the amount of MAGs used in the comparison, but we would argue that this leads to a more robust analysis of the different sequencing datasets.

A1.2.5 changes made: Edits was made to the caption of **Table 1** and **Fig S6** to provide clarification.

Q1.2.6: Firstly, in the Methods, it is actually quite hard to figure out what data has been incorporated in the binning procedures: can you simply list each of the analyses that are being included? (so the reader doesn't have to work so hard!). I presume that you are using the Illumina data to guide the binning, but that you are only considering the long read component of the bins in your analysis? Is that correct or not? In terms of the writing, it is also difficult to distinguish when you referring to the use of Illumina data for binning and when it is being used for sequence error correction.

A1.2.6 comment: We agree that the sheer number of datasets and comparisons included does make it complex.

To guide the binning of MAGs from the complex metagenomic sample, we used a time-series of Illumina data of the same anaerobic digester. The time-series data was mapped to the metagenomes to acquire contig coverage profiles, which were used as input for the binning. The time-series data was not used for short-read error-correction or assembly of any kind.

Only the Illumina MiSeq data (as described in Table 1), which was generated from the exact same digester sample as the long read datasets, was used for short-read polishing of the assembled metagenomes.

A1.2.6 changes made: We have made edits to the “Read assembly and binning” section to provide further clarification. We have also added a schematic, describing the main bioinformatics processes with complex metagenome data, to the supplementary material (**Figure S9**).

Q1.2.7: In Table 1 what does the “/+” mean in the second and third columns? Does this mean that in these results are from binning on the combined long read and short read derived contigs? What does “Contigs pr. HQ MAG (median)” actually mean? And why are there two numbers here? And are we meant to interpret them? Some clear, unambiguous table notes would be extremely useful here.

A1.2.7 comment: We apologize that this was indeed very unclear stated in the table. The “-/+” sign refers to the application of Illumina short-read polishing of the long-read metagenome assemblies. The way we intended to present the results was: unpolished/polished in order not to inflate the table with too many columns.

“Contigs pr. HQ MAG (median)” refer to the median number of contigs in HQ MAGs. This is also presented as before/after applying Illumina short read polishing (or unpolished/polished if keeping to the definition above).

A1.2.7 changes made: Additional table notes have been added to Table 1 to provide further clarification.

Q1.2.8: Table 1 has some key information missing, that is only discernable in Fig S6. For example, it appears as if the mean number of contigs per bin, from either PacBio or ONT, is around around 15 (I would guess based on the y-axis). This is critical information that should be included in Table 1, as it speaks directly to the overall recoverability of genomes.

A1.2.8 comment: We have provided the median number of contigs in HQ MAGs in Table 1 (see

A1.2.7). **A1.2.8 changes made:** A table note about the number of contigs in HQ MAGs was added to

Table 1.

Q1.2.9: In Table 1, why are the results for ONT-only not reported?

A1.2.9 comment: ONT-only results are reported in Table 1 (see **A1.2.7**). Again, we would like to apologize to the reviewer for the lack of clarity in Table 1.

A1.2.9 changes made: A table note explaining the distinction between Illumina unpolished/polished results was added to Table 1.

Q1.2.10: In Table 1, how many of these HQ-MAGs are single chromosomal sequences, whether circular otherwise?

A1.2.10 comment: We have added the single-contig HQ MAG counts, although we would like to note that conventional automated binners are not aware of contig circularity, and thus tend to bin circular contigs with linear ones.

A1.2.10 changes made: The number of single-contig HQ MAGs was added to Table 1.

Q1.2.11: In Fig S6A, I could not find results relating to combination of ONT+Illumina data. Given these seem to be reported in Table 1 [but see my comment above], why are they omitted here?

A1.2.11 comment: Since Illumina short read polishing does not cause breaks in the contigs and since we only used clustered MAGs in Fig S6A, there is no differences in metagenome fragmentation levels before/after applying short read polishing. Hence, we omitted the short read polished data from this plot to improve the clarity and readability.

A1.2.11 changes made: An extra note was added for Fig S6A to provide further clarification.

Q1.2.12: Fig S6B. Bulk genome quality statistics, why is contamination not being shown here? Does strain heterogeneity correlate with the polymorphic rate statistics?

A1.2.12 comment: Initially, we did have MAG contamination plots (see Fig S2 of our initial biorxiv submission: <https://www.biorxiv.org/content/10.1101/2021.10.27.466057v1.full.pdf>), but since then we omitted it as we did not see a significant difference in per-MAG contamination values from applying short read polishing. A plot for contamination values is also presented below:

22We have made a plot for clustered MAG SNP rates vs strain heterogeneity, although we did not observe any significant associations. The plot is presented below:

A1.2.12 changes made: MAG contamination values as well as the plot polymorphic rates against strain heterogeneity have been made available on the GitHub repository: <https://github.com/Serka-M/Digester-MultiSequencing/blob/main/code/plotting-mags/plotting-mags.md>

Q1.2.13: (line 17) “...can be used to generate near-perfect microbial genomes from isolates or metagenomes without short-read or reference polishing”. While that is technically correct, in the context of the real community analysis, one clearly still needs short read data to obtain the genomes in the first place (Table 1). It seems somewhat disingenuous to claim that for the consideration of local sequence quality, whether coding or non-coding, 10.4 sequencing obviates the need for Illumina sequencing, while ignoring the limitation that the underlying genomes would not, in all likelihood, be recovered without the use of the same data.

A1.2.13 comment: For producing the long-read metagenome assemblies, we assembled the long reads (Nanopore or PacBio) using the metaFlye assembler. Short Illumina reads were **not involved** in the long-read metagenome assembly.

We did short-read polishing on the long-read metagenomes as an additional step, in order to compare

genome quality with and without short-read polishing.

Furthermore, we used Illumina time-series data of the same digester plant from a previous unpublished project to assist with binning, as our goal was to acquire more MAGs so we could perform a more robust comparison. The Illumina time-series data can easily be substituted with long read time-series dataset, it's only a coverage measure.

A1.2.13 changes made: We have made edits to the Materials and methods section to provide further clarification on the usage of short reads in the study. We have also added a schematic, describing the main bioinformatics processes with complex metagenome data, to the supplementary material (**Figure S9**).

Decision Letter, first revision:

Subject: AIP Decision on Manuscript NMETH-BC47580A
Message:

Our ref: NMETH-BC47580A

12th Apr 2022

Dear Dr. Albertsen,

Thank you for submitting your revised manuscript "Oxford Nanopore R10.4 long-read sequencing enables near-finished bacterial genomes from pure cultures and metagenomes without short-read or reference polishing" (NMETH-BC47580A). It has now been seen by the original referees and their comments are below. The reviewers find that the paper has improved in revision, and therefore we'll be happy in principle to publish it in Nature Methods, pending minor revisions to satisfy the referees' final requests and to comply with our editorial and formatting guidelines.

TRANSPARENT PEER REVIEW

Nature Methods offers a transparent peer review option for new original research manuscripts submitted from 17th February 2021. We encourage increased transparency in peer review by publishing

25the reviewer comments, author rebuttal letters and editorial decision letters if the authors agree. Such peer review material is made available as a supplementary peer review file. Please state in the cover letter 'I wish to participate in transparent peer review' if you want to opt in, or 'I do not wish to participate in transparent peer review' if you don't. Failure to state your preference will result in delays in accepting your manuscript for publication.

Thank you again for your interest in Nature Methods Please do not hesitate to contact me if you have any questions.

Sincerely,

Lin Tang, PhD
Senior Editor
Nature Methods

ORCID

Reviewer #1 (Remarks to the Author):

The authors have addressed all of my previous points

Reviewer #2 (Remarks to the Author):

Thank you your detailed and considered responses to my questions, which are greatly appreciated. In particular, the issues about choice of terminology, importance of homopolymer analysis, the use of IDEEL statistics and the presentation of the AD data are all very convincing. Thank you for including Table S4 and Figure S9, which is very clear. My only additional request is that in Figure 2, the quoted coverage values are more precise, for improved clarity.

Author Rebuttal, first revision:

Reviewer #2:

Q2.1.1: My only additional request is that in Figure 2, the quoted coverage values are more precise, for improved clarity.

A2.1.1 changes made: Precise coverage values have been included in the figure legend.

Final Decision Letter:

Subject: Decision on Nature Methods submission NMETH-BC47580B
Message:

24th May 2022

Dear Dr Albertsen,

I am pleased to inform you that your Brief Communication, "Oxford Nanopore R10.4 long-read sequencing enables near-finished bacterial genomes from pure cultures and metagenomes without short-read or reference polishing", has now been accepted for publication in Nature Methods. Your paper is tentatively scheduled for publication in our July print issue, and will be published online prior to that. The received and accepted dates will be 10th Nov 2021 and 24th May 2022. This note is intended to let you know what to expect from us over the next month or so, and to let you know where to address any further questions.

27Your paper will now be copyedited to ensure that it conforms to Nature Methods style. Once proofs are generated, they will be sent to you electronically and you will be asked to send a corrected version within 24 hours. It is extremely important that you let us know now whether you will be difficult to contact over the next month. If this is the case, we ask that you send us the contact information (email, phone and fax) of someone who will be able to check the proofs and deal with any last-minute problems.

If, when you receive your proof, you cannot meet the deadline, please inform us at rjsproduction@springernature.com immediately.

Once your manuscript is typeset and you have completed the appropriate grant of rights, you will receive a link to your electronic proof via email with a request to make any corrections within 48 hours. If, when you receive your proof, you cannot meet this deadline, please inform us at rjsproduction@springernature.com immediately.

Once your paper has been scheduled for online publication, the Nature press office will be in touch to confirm the details.

Content is published online weekly on Mondays and Thursdays, and the embargo is set at 16:00 London time (GMT)/11:00 am US Eastern time (EST) on the day of publication. If you need to know the exact publication date or when the news embargo will be lifted, please contact our press office after you have submitted your proof corrections. Now is the time to inform your Public Relations or Press Office about your paper, as they might be interested in promoting its publication. This will allow them time to prepare an accurate and satisfactory press release. Include your manuscript tracking number NMETH-BC47580B and the name of the journal, which they will need when they contact our office.

About one week before your paper is published online, we shall be distributing a press release to news organizations worldwide, which may include details of your work. We are happy for your institution or funding agency to prepare its own press release, but it must mention the embargo date and Nature Methods. Our Press Office will contact you closer to the time of publication, but if you or your Press Office have any inquiries in the meantime, please contact press@nature.com.

Please note that *Nature Methods* is a Transformative Journal (TJ). Authors may publish their research with us through the traditional subscription access route or make their paper immediately open access through payment of an article-processing charge (APC). Authors will not be required to make a final decision about access to their article until it has been accepted. Find out more about Transformative Journals

To assist our authors in disseminating their research to the broader community, our SharedIt initiative provides you with a unique shareable link that will allow anyone (with or without a subscription) to read the published article. Recipients of the link with a subscription will also be able to download and print the PDF. As soon as your article is published, you will receive an automated email with your shareable link.

Please note that you and your coauthors may order reprints and single copies of the issue containing your article through Springer Nature Limited's reprint website, which is located at <http://www.nature.com/reprints/author-reprints.html>. If there are any questions about reprints please send an email to author-reprints@nature.com and someone will assist you.

Please feel free to contact me if you have questions about any of these points. Thank you very much again for publishing your paper at Nature Methods!

Best regards,

Lin Tang, PhD
Senior Editor

29